# Cell-free expression of the outer membrane protein OprF of *Pseudomonas aeruginosa* for vaccine purposes

Géraldine Mayeux[1] , Landry Gayet[1], Lavinia Liguori[1,2], Marine Odier[1,3], Donald K Martin[1], Sandra Cortès[4], Béatrice Schaack[1,5] , Jean-Luc Lenormand[1]

***Pseudomonas aeruginosa*** **is the second-leading cause of nosocomial infections and pneumonia in hospitals. Because of its extraordinary capacity for developing resistance to antibiotics, treating infections by** ***Pseudomonas*** **is becoming a challenge, lengthening hospital stays, and increasing medical costs and mortality. The outer membrane protein OprF is a well-conserved and immunogenic porin playing an important role in quorum sensing and in biofilm formation. Here, we used a bacterial cell-free expression system to reconstitute OprF under its native forms in liposomes and we demonstrated that the resulting OprF proteoliposomes can be used as a fully functional recombinant vaccine against** ***P. aeruginosa*.** **Remarkably, we showed that our system promotes the folding of OprF into its active open oligomerized state as well as the formation of mega-pores. Our approach thus represents an easy and efficient way for producing bacterial membrane antigens exposing native epitopes for vaccine purposes.**

## Introduction

The Gram-negative bacterium *Pseudomonas aeruginosa* (*P. aeruginosa*) is a ubiquitous and opportunistic pathogen causing health care–associated infections such as central line–associated bloodstream infections, ventilator-associated pneumonia, catheter-associated urinary tract infections, and surgical-site infections, which can be invasive and lethal for patients having weakened immune defenses (Gaynes & Edwards, 2005). *P. aeruginosa* also frequently colonizes the respiratory tract of patients suffering of diverse obstructive lung diseases such as cystic fibrosis (CF), and the resulting chronic infections are a major leading cause of morbidity and mortality in these patients (Emerson et al, 2002; Valderrey et al, 2010). *P. aeruginosa* being now resistant to many commonly used antibiotics, there is an urgent need for having available new antimicrobial strategies to be able to counteract these infections (Poole, 2011; Tacconelli, 2017). Because of the

remarkable capacity of *P. aeruginosa* to develop or acquire new resistance mechanisms to new antibiotics, preventing infections through vaccination represents a reasonable and promising approach to overcome *P. aeruginosa* antibiotic resistance (Pang et al, 2019; Baker et al, 2020).

The outer membrane proteins (OMPs) of Gram-negative bacteria represent ideal vaccine candidates because most of them expose epitopes on bacterial surface that can be recognized by the host immune system. Among these OMPs, Oprs proteins are highly conserved and antigenically related among all serotypes of *P. aeruginosa* making them good immunogens for vaccine development (Chevalier et al, 2017). OprF is the most abundant non-lipoprotein OMP in *P. aeruginosa*, and it has been shown that its amino acid sequences were highly conserved in all 134 pathogenic and environmental strains of *P. aeruginosa* (Moghaddam et al, 2017).

OprF was described as a closed and as an open channel conformer. In its closed conformation, the N-terminal part of OprF (aa 1–162), which represents half of the protein, spans eight times through the membrane forming an eight stranded β-barrel (Brinkman et al, 2000) whereas the C-terminal part of OprF (aa 210–326) is located in the periplasm as a globular domain composed of α-helices and/or β-strands (Sugawara et al, 2012). In its open conformation, the C-terminal part is located in the outer membrane forming with the N-terminal part an integral membrane protein characterized by 14–16 transmembrane passages (Sugawara et al, 2006, 2012). The closed channel conformation is the most represented (≈95%) and is thought to play a structural role for the bacterial cell wall. In this conformation, the C-terminal peptidoglycan-binding domain anchors the outer membrane to the peptidoglycan layer (Woodruff & Hancock, 1989; Rawling et al, 1998). By contrast, the open channel conformer is rare (<5% of all the conformers) and its role remains undefined (Sugawara et al, 2006). Other functions have also been assigned to OprF such as biofilm formation, outer membrane vesicles biogenesis, binding and adhesion to host cells, involvement in quorum-sensing response, and sensor of host immune activation through IFN-γ binding, all related

[1]TheREx and Synabi, University Grenoble Alpes, CNRS, Centre Hospitalier Universitaire Grenoble Alpes, Grenoble Institut Polytechnique (INP), Translational Innovation in Medicine and Complexity (TIMC), Grenoble, France   [2]Maison Familiale Rurale Moirans, Moirans, France   [3]Catalent Pharma Solutions, Eberbach, Germany   [4]Synthelis, La Tronche, France   [5]University Grenoble Alpes, Commissariat à l'Énergie Atomique et aux Énergies Alternatives (CEA), CNRS, Institut de Biologie Structurale (IBS), Grenoble, France

Correspondence: jllenormand@chu-grenoble.fr

to *P. aeruginosa* virulence (Azghani et al, 2002; Wu et al, 2005; Fito-Boncompte et al, 2011; Wessel et al, 2013; Alhede et al, 2014). It remains to be determined to what extent these functions depend on the open channel conformer.

In 1995, von Specht et al (1995) were the first to demonstrate that immunization with a recombinant hybrid protein consisting of the GST-linked C-terminal part of OprF (aa 190–342) fused to another Opr protein, OprI (aa 21–83) expressed in *Escherichia coli*, protected mice against a lethal *P. aeruginosa* infection (von Specht et al, 1995). The use of GST being not accepted as part as a recombinant vaccine candidate in humans, the hybrid protein OprF/OprI was then expressed in *E. coli* and purified under native conditions as the histidine-tagged fusion protein Met-Ala-(His)$_6$OprF$_{190-342}$-OprI$_{21-83}$ (Mansouri et al, 2003; von Specht et al, 2000). An initial randomized placebo-controlled phase II clinical study performed in ventilated intensive care unit patients confirmed the immunogenicity and the safety of this vaccine, called IC43. Because this phase II study was not designed to assess the protective effect of the vaccine, a confirmatory multicenter and double-blinded phase II/III clinical study was conducted and concluded that there was no clinical benefit of IC43 in terms of overall mortality compared to placebo treatment (Adlbrecht et al, 2020).

Developing a vaccine against a membrane protein remains a big challenge, especially if the target is a multispanning membrane protein such as OprF. One of the most critical prerequisite for successful vaccine development is the production of the membrane antigen on a large scale and in its native conformation. Membrane antigen reconstitution in proteoliposome for subsequent immunization is a valuable strategy to induce a specific and high quality humoral response against all the natural and conformational epitopes of the membrane target (Hashimoto et al, 2018). However, overexpression of a correctly folded recombinant membrane protein in cellular system for subsequent reconstitution in proteoliposome can be difficult. Many integral membrane proteins, including OprF, exhibit cellular toxicity on induction and give low yields of correctly folded recombinant protein (Duchêne et al, 1988).

Cell-free protein synthesis (CFPS) system offers considerable advantages in the context of vaccine development against membrane proteins. Compared with cell-based systems, membrane proteins expression in cell-free systems cannot be impaired by their potential lethality (Kimura-Soyema et al, 2014). Some of these CFPS systems, such as those based on *E. coli* or wheat germ extracts, can also be easily adapted for large-scale protein production as for industrial applications (Swartz, 2006; Beebe et al, 2011; Zawada et al, 2011). In the presence of liposomes or nanodiscs, cell-free systems allow the membrane protein to be translated and spontaneously inserted into the liposomal membrane in a one-step reaction, skipping the laborious solubilization, purification, and proteoliposome reconstitution steps required when using cellular expression systems (Liguori et al, 2016, 2015; Liguori and Lenormand, 2009; Sachse et al, 2014). Finally, CFPS systems also enable rapid screening of different liposomal compositions, therefore increasing the likelihood of reconstituting the membrane entire protein in its active form (Kimura-Soyema et al, 2014) and presenting native antigens (He et al, 2017).

Previously, we reported the reconstitution of the full-length OprF protein of *P. aeruginosa* in proteoliposomes by using an optimized *E. coli*–based cell-free expression system in the presence of synthetic liposomes. We demonstrated by using neutron reflectometry and impedance spectrometry that fusion of those recombinant proteoliposomes into preformed tethered lipid bilayer membranes resulted in the incorporation of OprF, at least in its native closed conformation (Maccarini et al, 2017). In this study, we undertook the biochemical and biophysical characterization of these OprF recombinant proteoliposomes. We discovered that the system of in vitro protein expression in the presence of liposomes of a specific lipid composition promoted the reconstitution of OprF not only in its native closed conformation but also, for a large proportion, in its rare active open conformation, resulting in the formation of mega-pores formation across the liposomal membrane. We then for the first time investigated the vaccine potential of OprF full-length protein in proteoliposomes, exposing all the natural conformational epitopes of OprF, in a murine model of *P. aeruginosa* acute pulmonary infection. Notably, we succeeded in protecting mice from a mucoid strain against which immunization had never been achieved using previous partial misfolded OprF vaccine (Hassan et al, 2018).

# Results

### Part 1: OprF proteoliposomes production

#### *Influence of liposome composition and concentration on cell-free OprF production*

Using a bacterial lysate, OprF was expressed as a His$_6$-tagged fusion protein in presence of liposomes of six different lipid compositions (LC 1–3′) to determine whether these lipid compositions had an influence on OprF cell-free expression yield. LC 1 (cholesterol, DOPC, DOPE, and DMPA; 2-4-2-2) was used as a reference because, according to Maccarini et al (2017), it provided the highest yield of expressed OprF protein (Maccarini et al, 2017). LC 2 and 3 contained both POPE, POPG, and *E. coli* CL lipids mimicking *P. aeruginosa* plasma membrane composition (Ghorbal et al, 2013; Benamara et al, 2014). DMPA was added in LC 3 and MPLA was added in LC 1′, 2′, and 3′ as it could be used as a vaccine adjuvant (Alving et al, 2012) (Table 1). Western blot analysis was performed to evaluate OprF expression levels, and the results did not show any significant OprF expression level differences between lipid compositions (Fig 1A). The influence of liposome concentration on OprF synthesis was then tested for each lipid composition. Fig 1B shows that the production of OprF in presence of an increasing concentration of liposome (1–4 mg/ml) did not significantly increase OprF expression yield for every lipid composition. Therefore, the reference LC1 at 4 mg/ml was chosen to produce OprF protein with the cell-free system.

#### *OprF proteoliposomes purification*

After cell-free OprF production in the presence of liposomes, OprF proteoliposomes were separated from aggregated OprF proteins and empty liposomes, using a sucrose density gradient centrifugation. The cell-free reaction mixture was loaded onto the top of a 0–40% sucrose gradient and centrifuged at 287,660$g$ for 2 h at 4°C. After

**Table 1.   Content of the different lipid compositions.**

| Lipid composition (LC) | Lipids | Molar ratio |
|---|---|---|
| 1 | Cholesterol | 2-4-2-2 |
| | 1,2-dioleoyl-sn-glycero-3-phosphocholine (DOPC) | |
| | 1,2-dioleoyl-sn-glycero-3-phosphoethanolamine (DOPE) | |
| | 1,2-dimyristoyl-sn-glycero-3-phosphate (sodium salt) (DMPA) | |
| 1' | LC1 + 1 mg/ml monophosphoryl lipid A (MPLA) | |
| 2 | 1-palmitoyl-2-oleoyl-sn-glycero-3-phosphoethanolamine (POPE) | 6-2-2 |
| | 1-palmitoyl-2-oleoyl-sn-glycero-3-phospho-(1'-rac-glycerol) (sodium salt) (POPG) | |
| | *E. coli* cardiolipin (CL) | |
| 2' | LC2 + 1 mg/ml MPLA | |
| 3 | POPE, POPG, *E. coli* CL, DMPA | 6-2-1-1 |
| 3' | LC3 + 1 mg/ml MPLA | |

centrifugation, liposomes floated on the surface whereas recombinant proteoliposomes were detected as separate bands in the gradient. 1-ml fractions were recovered from the top to the bottom of the gradient and were analyzed by Western blot. OprF proteoliposomes were detected in the gradient layers containing 10–20% sucrose. As shown in Fig 1C, a band corresponding to the monomeric full length OprF protein (41 kD) appeared between molecular weight markers 37 and 50 kD, whereas bands corresponding to OprF oligomers (dimer, trimer, and tetramer) non-aggregated and resistant to denaturation appeared between molecular weight markers 75 kD and above 150 kD. The band around the molecular weight marker 37 kD corresponded probably to the product of an OprF aborted synthesis. Unincorporated OprF protein was not detected at the bottom of the gradient (Fig 1C). Sucrose gradient layers containing OprF proteoliposomes were then centrifuged at 100,000*g* for 30 min to pellet proteoliposomes. The pellet was washed twice with NaCl 5M to remove protein contaminants bound to proteoliposomes (Maccarini et al, 2017). After resuspension of the pellet in Tris buffer, the purity of OprF proteoliposomes was evaluated on a Coomassie-stained SDS–PAGE gel. Two major bands were easily detected, one corresponding to full length OprF and the other one to its premature termination of translation. Very thin bands were also detected between molecular weight markers 50 and 250 kD (Fig 1D). These bands could correspond either to different oligomeric forms of OprF, as observed by Western blot, or to a few protein contaminants (≈5%) (Fig 1C and D). Taken together, these results indicated that cell-free expression system in the presence of liposome is a powerful strategy for obtaining high levels of pure (up to 95%) recombinant OprF proteoliposomes, with a yield ranging from 0.5 to 1 mg/ml.

### Part 2: biochemical and biophysical characterization of OprF proteoliposomes

#### *Determination of the OprF orientation into the lipid bilayer by Atomic Force Microscopy (AFM)*

One of the major concerns regarding the expression of membrane proteins using a cell-free expression system and their incorporation into liposomes resides in the protein orientation into the resulting proteoliposome. To determine the orientation of the OprF membrane protein into the liposome, we first analyzed pure recombinant proteoliposomes by AFM. Samples of OprF proteoliposomes were absorbed on a mica surface and analyzed by AFM using a Tris–NTA–modified tip binding the N-terminal 6xHis-tag of OprF. AFM analysis was performed with and without Triton X-100 detergent. A Triton X-100 solution was used to solubilize OprF proteins from the liposomal membrane, thereby exposing all the His-tag located inside the liposome and allowing them to be bound by the Tris–nitrilotriacetate (NTA)–modified tip. Topographic images, acquired without (Fig 2A and B) and with Triton X-100 (Fig 2C), highlighted OprF proteoliposomes on the mica surface (circles in white dotted lines). In the absence of Triton X-100, very few specific adhesion events between the Tris–NTA modified tip and the N-terminal His-tag of OprF occurred on the surface of proteoliposomes, as shown on the corresponding adhesion maps illustrating adhesion forces between 80 and 150 pN (Fig 2A' and B'). In contrast, in presence of Triton X-100, many specific adhesion events were detected (Fig 2C'). On average, the Tris–NTA modified tip bound the His-tag of one OprF protein out of six without Triton, and of five OprF proteins out of six with Triton, demonstrating that the N-terminal 6xHis-tag of OprF was mainly located inside the liposome. This result is in accordance with the model of OprF protein proposed by Sugarawa and colleagues in which the N-terminal extremity of OprF points towards the perisplasmic space (Sugawara et al, 2012).

#### *Topology determination of OprF proteins into the liposomal membrane using trypsin digestion and AFM*

OprF proteoliposomes purified by sucrose gradient density centrifugation were submitted to a limited proteolysis experiment to determine the topology of OprF in the liposomal membrane. OprF protein sequence contains 32 trypsin cleavage sites. Without membrane protection of OprF, trypsin would generate peptides with a mass ranging from 146 to 4,649 D, as computed by the

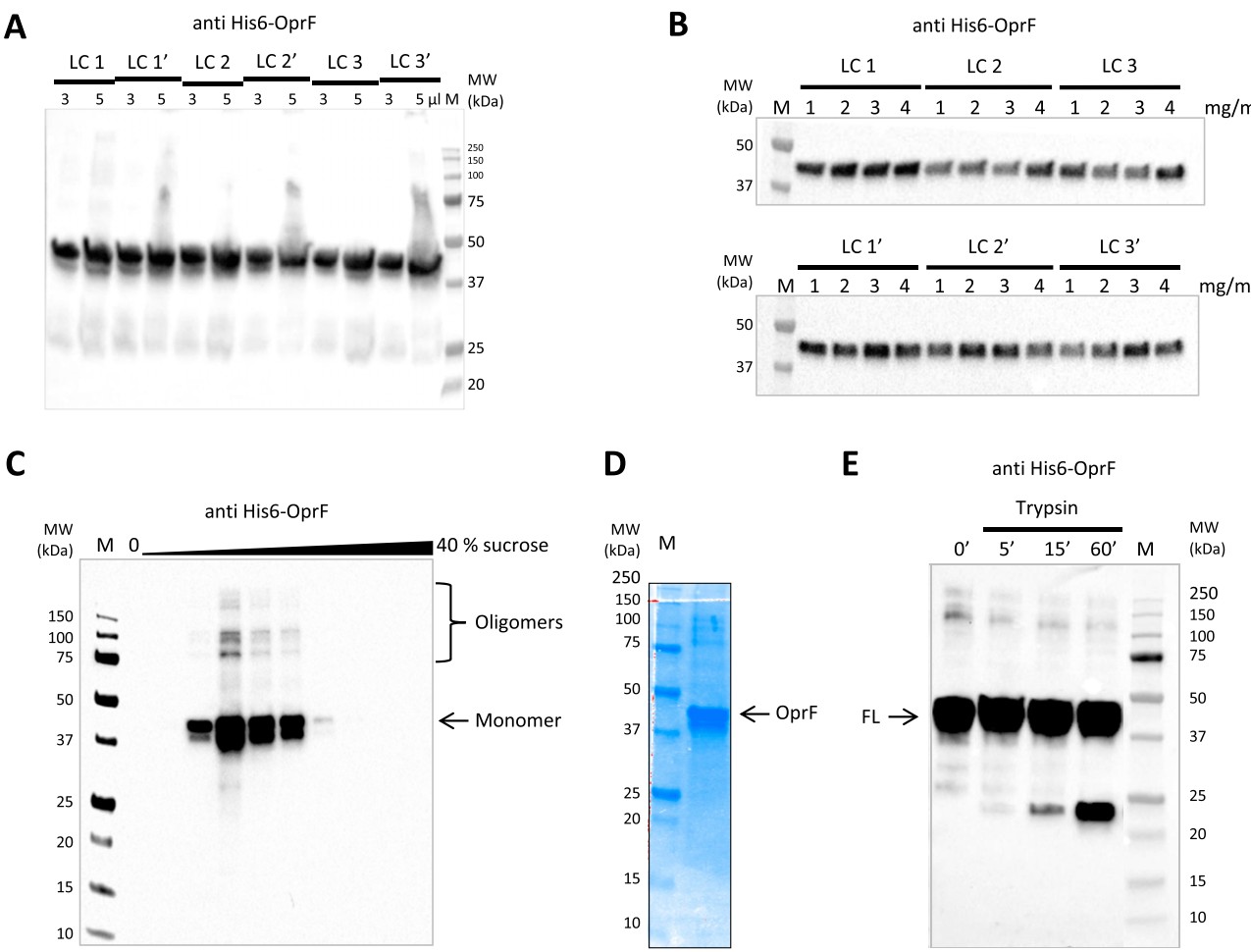

**Figure 1.  Production and purification of recombinant OprF proteoliposomes.**
**(A)** Comparison, using anti-His Western blot analysis, of cell-free OprF expression in the presence of liposomes of six different lipid compositions (LC 1 to LC 3′) 3 and 5 µl of each cell-free reaction mixture were analyzed. Anti-His antibody was conjugated to HRP and was detected using chemiluminescence. **(B)** Comparison, using anti-His Western blot analysis, of the influence of liposome concentration on OprF expression for each LC. **(C)** Purification of OprF proteoliposomes (LC 1) by sucrose gradient. Anti-His Western blot analysis of each 1 ml layer of the sucrose gradient (10 lanes from 0 to 40% sucrose), with OprF monomeric and oligomeric forms indicated (only dimeric, trimeric, and tetrameric forms are visible). **(D)** Evaluation, using Coomassie gel, of the purity of OprF proteoliposomes after purification by sucrose gradient and subsequent pellet washing with NaCl. **(E)** Trypsin digestion of OprF proteoliposomes after purification. The digested bands were revealed by Western blotting using an anti-his antibody.

PeptideCutter program. The result of the trypsin digestion of OprF membrane protected proteoliposomes, analyzed by Western blot using an anti-His antibody, demonstrated that OprF adopts at least two different membrane topologies in liposomes. One in which OprF is fully embedded into the membrane and thus protected from proteolysis because the signal corresponding to full-length His-tagged OprF (FL) did not disappear over time. And another in which only approximately half of the His-tagged protein is embedded into the membrane because a smaller protein fragment located between molecular weight 20 and 25 kD was generated over time (Fig 3).

These first observations were then corroborated and refined using AFM. The analysis of force distance (FD) curves showing specific adhesion events in Triton X-100 solution indicated that OprF adopts two different transmembrane topologies in the liposomal membrane corresponding to its closed and open channel conformer. Based on the Worm-Like Chain model, 64% of specific adhesion events

corresponded to eight transmembrane domains (closed channel conformer) (Fig 4A) and 36% of specific adhesion events corresponded to 16 transmembrane domains (open channel conformer) (Fig 4B).

### Prediction of OprF secondary structure in proteoliposomes using circular dichroism (CD) spectroscopy

CD experiments were performed to determine OprF secondary structure in liposomes (LC1) as well as the influence of IFN-γ addition to it (Fig 5). CD spectrum obtained before IFN-γ addition showed three negative bands, at 212, 215, and 223 nm, as well as a positive band at 202 nm. Using the Web server BeStSel, an estimation of OprF secondary structure based on this spectrum found 14% α helix, 36% β strand, 15% β turn, and 36% disordered structures, which is close to the secondary structure prediction deduced by Sugawara et al (1996) from the CD spectra acquired with OprF in detergent (Sugawara et al, 1996). CD spectrum recorded after IFN-γ addition

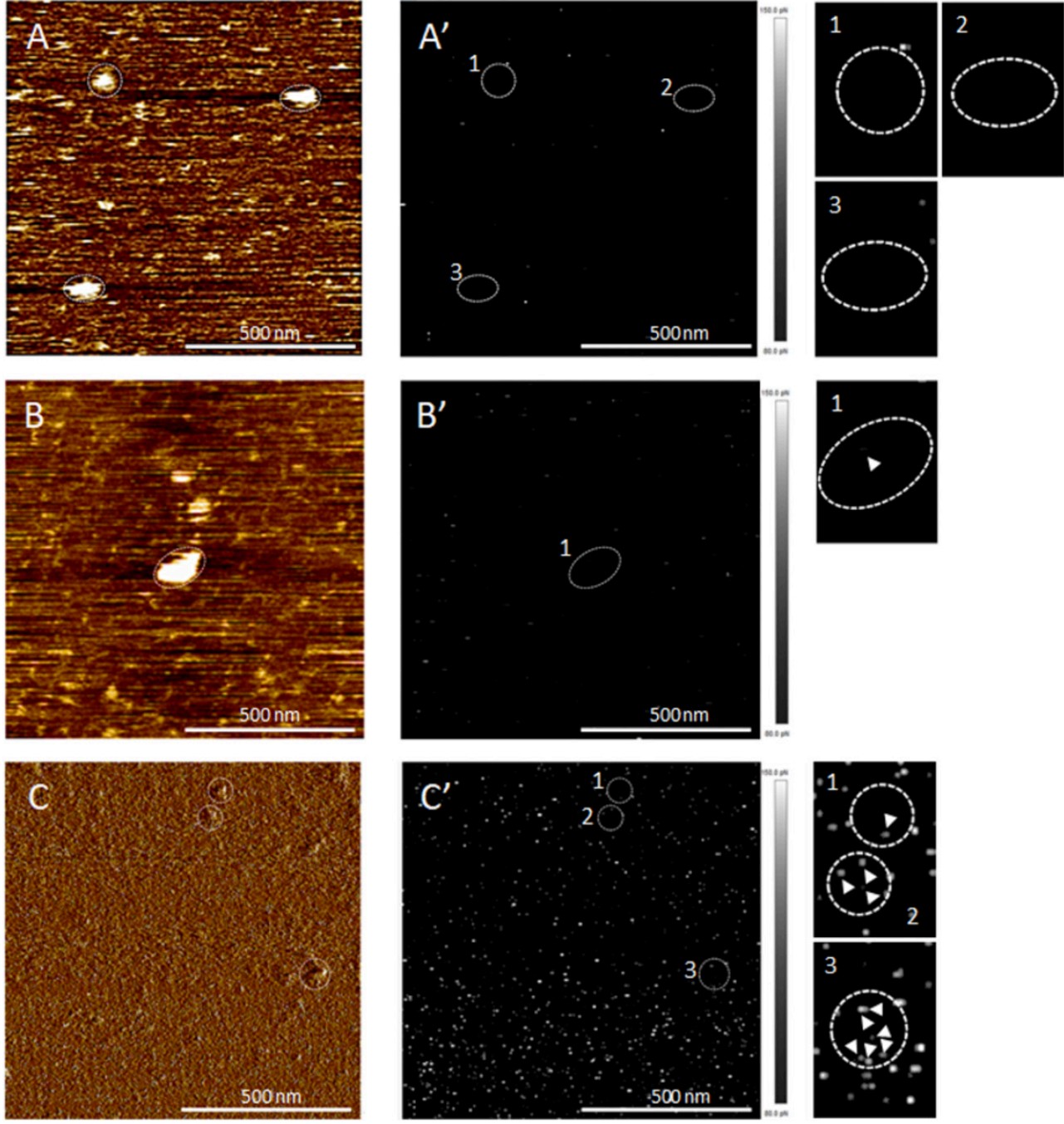

**Figure 2. Determination of OprF orientation in proteoliposomes by atomic force microscopy.**
**(A, B, C)** Atomic force microscopy topographic images of an OprF proteoliposomes' sample absorbed on mica surface, acquired using a tip functionalized with Tris–NTA, without (A, B) and with Triton X-100 (C). The circles in white dotted lines correspond to proteoliposomes. **(A′, B′, C′)** Corresponding adhesion maps of specific adhesion forces between the Tris–NTA modified tip and OprF His-tag, without (A′, B′) and with Triton X-100 (C′). The white to grey squares represent values of adhesion forces between 80 and 150 pN, as illustrated by the scale bar on the right. The circles in white dotted lines corresponding to proteoliposomes are zoomed on the right. The white arrows indicate specific adhesion points validated by the force distance curves and correspond to 16 transmembrane passages based on the theoretical model Worm-Like Chain.

showed also three negative bands, at 214, 220 and 224 nm, but their values were shifted to the right, whereas the positive band value, at 199 nm, shifted to the left. In consequence, the prediction of OprF secondary structure changed to 18% α helix, 26% β strand, 13% β turn, and 43% disordered structures by the addition of IFN-γ (Fig 5).

**Investigation of OprF pore-forming activity in proteoliposomes using negative staining electron microscopy and AFM**
Negative staining electron microscopy and AFM analysis of OprF proteoliposomes were then used to visualize OprF pore forming in the liposomal membrane. On electron microscopy images, a series

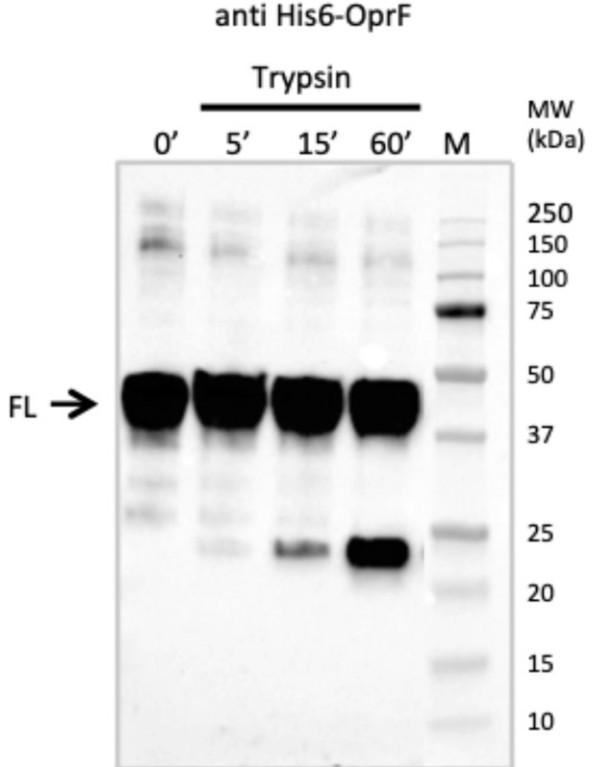

**Figure 3. Determination of the membrane topology of OprF in liposomes by trypsin digestion.**
The result of the trypsin digestion of OprF at different incubation times is visualized by anti-His Western blot (FL, full length). Anti-His antibody was conjugated to HRP and was detected using chemiluminescence.

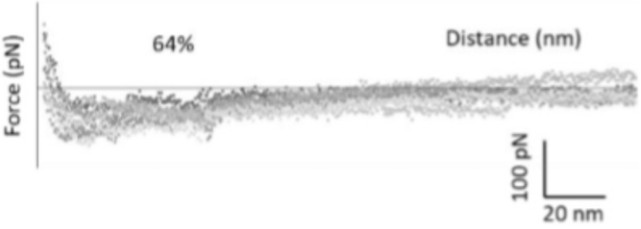

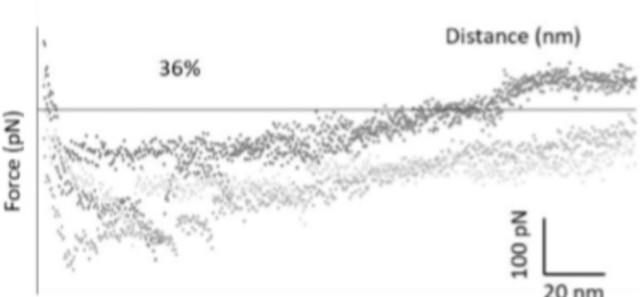

**Figure 4. Detection of two transmembrane topologies adopted by OprF in proteoliposomes by Atomic Force Microscopy.**
**(A, B)** Force distance curves, obtained with a tip functionalized with Tris–NTA in Triton X-100 solution, detecting 64% of specific adhesion events corresponding to eight transmembrane domains (A) and 36% of specific adhesion events corresponding to 16 transmembrane domains (B), based on the theoretical model Worm-Like Chain.

of holes with an average size of 9.5 ± 4 nm (mean ± SD) corresponding to pores were observed through the membranes of OprF proteoliposomes (Fig 6B–E). Such liposomal membrane perforation was not observed on the images of liposomes incubated with the cell-free system's cell lysate and reaction mixture without DNA (negative control) (Fig 6A). Besides, AFM topographic images of the surface of OprF proteoliposomes also revealed the presence of pores surrounded by OprF proteins and having an average diameter of 10 nm (Fig 7A–C). These data suggest that the pore forming was assigned to OprF protein activity in the liposomal membrane.

### Assessment of OprF ion channel activity and influence of IFN-γ

Then, to evaluate the ion channel activity of OprF proteins reconstituted into proteoliposomes as well as the effect of IFN-γ on it, the steady-state conductance of a tethered bilayer lipid membranes (t-BLM) was measured using the Tethapod after the fusion of recombinant OprF proteoliposomes or empty liposomes (control experiment) to the t-BLM promoted by DDM. The TethaPod system used an alternating-current impedance spectroscopy technique that was adapted to the tethered lipid bilayer so as to provide a measurement of the conductance of pore-forming proteins. We have already shown that such impedance spectroscopy provides a precise measurement of the ion flux across the membrane driven by the OprF porin (Maccarini et al, 2017). In Fig S1A, evidence that the OprF operates as an ion channel was provided by the

measured conductance of the membrane increasing in response to increases in the concentration of KCl on one side of the t-BLM. The incorporation of OprF in the t-BLM resulted in a steady-state conductance measured at around 0.8 $\mu$S for a 150 mM KCl concentration. The conductance was greatest with 1M KCl (1.1 $\mu$S) in the PL OprF t-BLM when compared with one of the empty liposome t-BLM (0.3 $\mu$S). The fact that this last value was not zero was due to the small permeation of the membrane by the DDM detergent. Furthermore, we also tested the inhibitory effect of IFN-γ. In Fig S1B, we showed first that the OprF provided a conducting pore to the membrane when added to the t-BLM with a conductance of 1.04 $\mu$S in the presence of 150 mM KCl. Then, this K$^+$ permeability was decreased by the addition of increasing concentrations of INF-γ (0.3–1.5 $\mu$M) into the well, reducing the conductance from 1.04 to 0.3 $\mu$S. Again a small permeability could be detected in the control t-BLM as a result of the presence of DDM. The electrical impedance spectroscopy measurements were adjusted using the tethaquick software model where the parameters were the capacitance of the t-BLM (Cm), the resistance of the membrane to account for the OprF porin (Rm), and a constant phase element to account for the impedance of the tether region below the t-BLM (CPEteth).

### Part 3: OprF proteoliposomes as a vaccine strategy

### Immunization of mice using recombinant OprF proteoliposome

To assess the possibility of using the recombinant OprF proteoliposome as a vaccine, mice were immunized with different

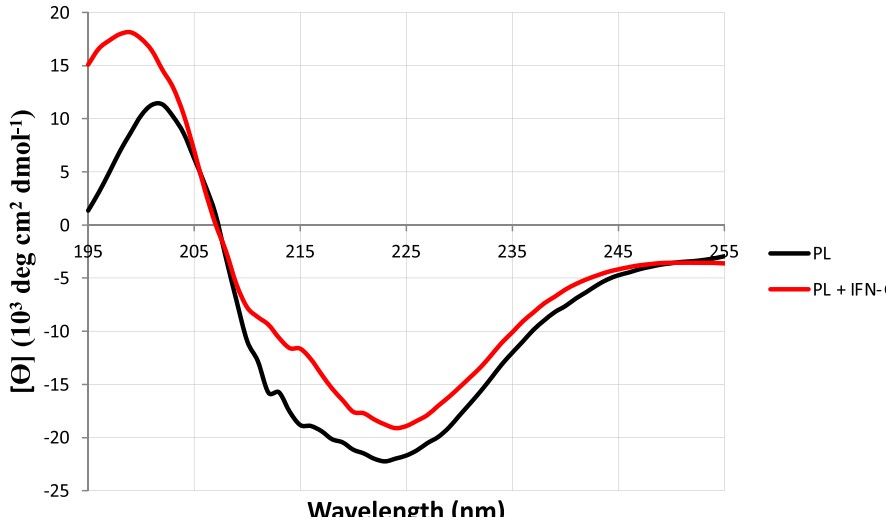

**Figure 5.** Superimposition of circular dichroism spectra of OprF proteoliposomes before (black curve) and after addition of IFN-γ (red curve).

concentrations of OprF proteoliposome followed by an infectious challenge with a lethal dose (2 × 10$^7$ CFU) of the mucoid CF isolate CHA pathogenic strain of *P. aeruginosa* 1 wk after the fourth immunization (Bezuidt et al, 2013). Mice were monitored for specific clinical signs including weight loss and behavior and survival for 80 h. As shown in Fig 8A and B, mice infected with empty liposomes, which were previously incubated with the cell-free lysate before injection, died within 42 h after the

bacterial challenge. In contrast, mice vaccinated with different doses of OprF proteoliposomes showed a better survival rate (90%) and no longer showed apparent traces of disease, recovering normal activity within 80 h after the bacterial challenge (Fig 8A). To ensure the reproducibility of the vaccination scheme, a new vaccination trial was performed using fresh OprF proteoliposomes, and a second batch of mice. Again, we found that only vaccinated mice were protected against a lethal dose of the CHA strain and no signs of disease were detectable

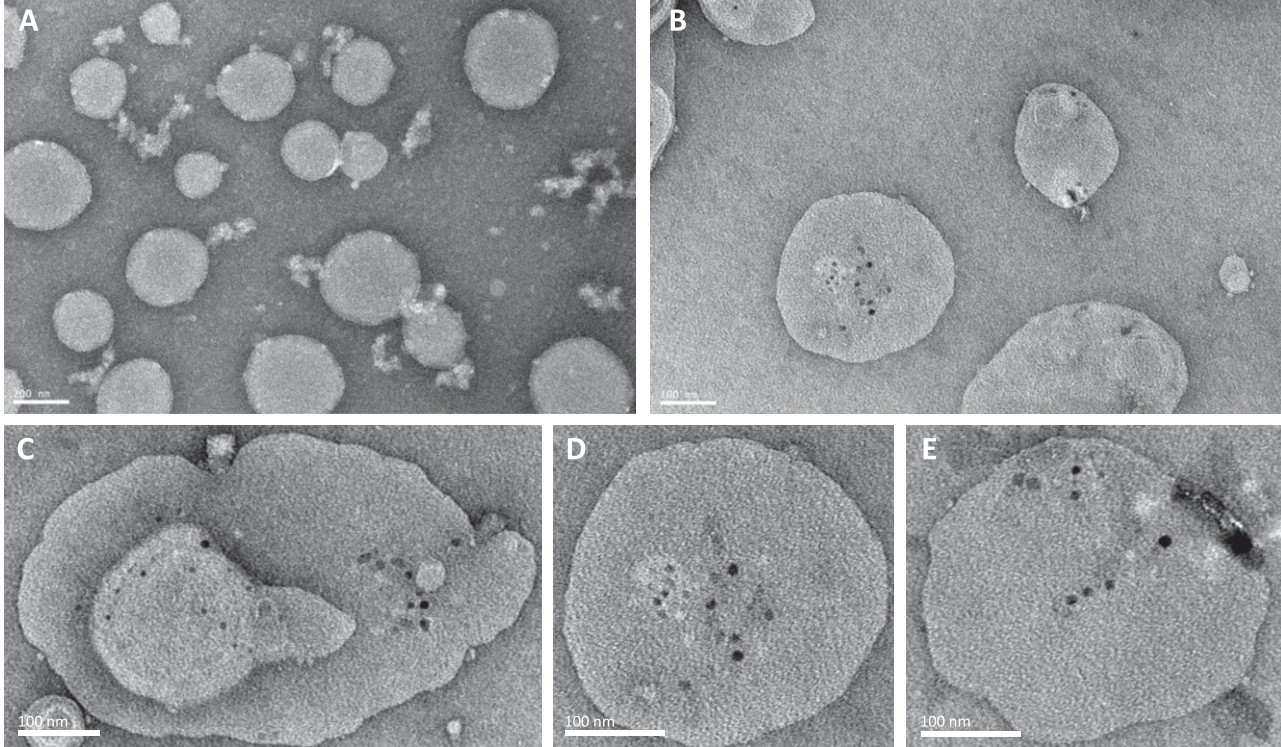

**Figure 6. Visualization of liposomal surface by negative-staining (NS) EM. Scale bar 100 nm.**
**(A)** EM image of empty liposomes (incubated using the cell-free cell lysate and reaction mixture = negative control). **(B, C, D, E)** EM images of OprF proteoliposomes (LC1). The membrane was perforated with a series of holes with an average size of 9.5 ± 4 nm corresponding to OprF pores.

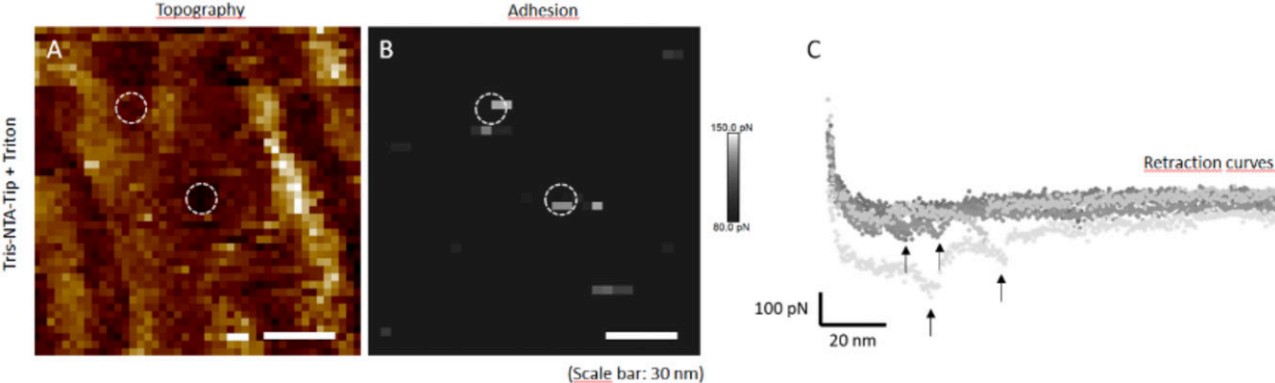

**Figure 7. Visualization of OprF pore-forming activity by Atomic Force Microscopy.**
**(A)** Topographic image of the surface of an OprF proteoliposome absorbed onto a freshly cleaved mica support, acquired using a tip functionalized with Tris–NTA, in presence of Triton X-100. **(B)** Corresponding adhesion map of specific adhesion forces between the tris–NTA modified tip and OprF His-tag. The white to grey squares represent adhesion forces values between 80 and 150 pN, as illustrated by the scale bar on the right. The circles in white dotted lines highlight pores. **(C)** Force distance curves detecting specific adhesion events (indicated by arrows).

after 80 h (Fig S2). These data suggested that immunization using OprF proteoliposomes produced with a cell-free based system represented a strong vaccine vector candidate for potential uses in humans.

### Evaluation of the humoral response after immunization with recombinant OprF proteoliposomes

Sera from mice immunized with 25, 50, or 75 µg of recombinant OprF proteoliposome were obtained from retro-orbital samples 1 wk after the fourth immunization (day 48). These sera were then analyzed by indirect ELISA for determining the titer of anti-OprF antibodies. The superimposition of the different titration curves showed that the IgG titers (1/160,000) in mice sera did not change in relation to the quantity of OprF proteoliposome used for immunization (Fig 9).

In addition, Western blotting using these sera confirmed the presence of polyclonal antibodies capable to bind either the OprF recombinant protein reconstituted in proteoliposomes using the cell-free system or the OprF protein expressed by *P. aeruginosa* CHA strain (Fig 10).

### Assessment of the cross-protection conferred by sera from immunized mice

To assess whether the immune response contributes to the cross-protection against a *P. aeruginosa* infection, naïve mice were challenged with a lethal dose of the CHA strain and passive

injection with sera containing polyclonal antibodies induced by the OprF proteoliposomes was performed 1 h after the bacterial challenge. We found that the passive injection of naïve mice with the sera from immunized mice resulted in a total protection against infection by the CHA strain indicating that the vaccination produced neutralizing antibodies (Fig 8B). Taken together, these results indicate that the vaccination strategy using an OprF proteoliposome synthesized by the cell-free expression system was able to induce a humoral response by producing neutralizing antibodies and to prevent or cure *P. aeruginosa* infections.

### Determination of the storage stability of OprF proteoliposomes

To confirm that OprF proteoliposomes could be stored for vaccine uses, integrity of OprF was evaluated after proteoliposomes storage at RT, 4°C, –20°C, and –80°C for 2 wk. Analysis by SDS–PAGE and Western blotting showed that OprF integrity was preserved for 14 d when proteoliposomes were stored at 4°C, –20°C, and –80°C because no protein degradation was observed. Full-length OprF as well as its oligomeric forms ranging from 75 kD to more than 250 kD were preserved. In contrast, OprF appeared to be degraded after 2 wk at RT. The His-tag was not detected by Western blot and when compared to the control sample, the band corresponding to full length OprF on SDS–PAGE gel disappeared and new bands of lower

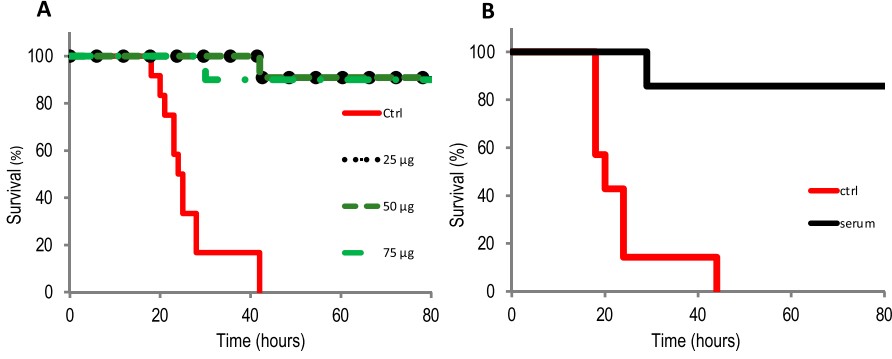

**Figure 8. Immunization with recombinant OprF proteoliposomes protects mice against a bacterial challenge.**
**(A)** C57Bl/6 mice were immunized subcutaneously with 25, 50 or 75 µg of OprF in proteoliposome. 1 wk after the fourth immunization, OprF-immunized groups (n = 10 or 11) or control group (n = 11) were subcutaneously challenged with a lethal dosage of *Pseudomonas aeruginosa* CHA strain (2 × $10^7$ CFU/mouse) and monitored for survival and disease symptoms for up to 80 h. **(B)** Naïve mice were subcutaneously challenged with a lethal dose of the CHA strain and an intraperitoneal injection with sera containing polyclonal antibodies induced by the OprF proteoliposomes was performed 1 h after the bacterial challenge. Mice were monitored for survival and disease symptoms for up to 80 h.

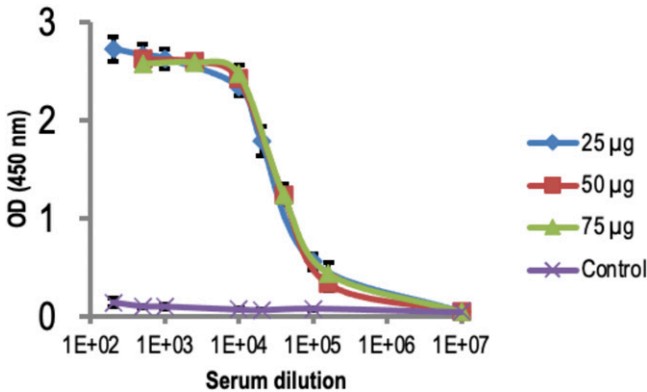

**Figure 9. Anti-OprF antibody titration curves of sera from immunized mice.**
Antibody titration curves of sera from mice immunized four times with 25, 50, or 75 µg of recombinant OprF proteoliposome. Control mice were immunized with liposomes previously incubated with cell-free lysate. Values are means of three mice per group determined by indirect ELISA using recombinant OprF proteoliposome as coated antigen.

molecular weight appeared (Fig 11A and B). Storage time of liposomes depends on several factors, including temperature, pH, and medium. Lipid hydrolysis begins immediately after resuspension of the lipid film in buffer and is dramatically increased by temperature. Hydrolysis products such as free fatty acids and monoacyl derivatives (lyso lipids), acting as detergents, permeabilize the membrane. In general, after 5–7 d at 4–8°C and pH 7.4, liposomes will start to display structural changes and begin to leak. In consequence, at RT, liposomes integrity was rapidly compromised and they could no longer ensure OprF protection from proteolysis leading to OprF degradation as observed after 2 wk.

## Discussion

Multiple *P. aeruginosa* targets have already been used as potential protective antigens for vaccine development. Those include LPS, alginate, flagella, exotoxins, and proteins from the type 3 secretion system (PopB and PcrV) and OMPs (OprF and OprI). However, most of the vaccine strategies based on these antigens failed when tested in animal models or in clinical trials because of either a lack of efficacy or the variability of the bacterial target in the different serotypes.

So far, two different approaches have been used to develop a protein-based vaccine targeting either the C-terminal or the N-terminal part of OprF. The first one was based on the direct administration of a recombinant hybrid protein composed of the C-terminal part of OprF (aa 190–342) fused to OprI (aa 21–83) (von Specht et al, 1995). This IC43 vaccine demonstrated some interesting immune response in mice and was safe and well tolerated after administration to human. However, it did not achieve the primary goal of clinically meaningful reduction of every cause of mortality (Adlbrecht et al, 2020). Several surface-exposed B-cell epitopes have been localized in the carboxy-terminal sequence of OprF (aa 190–342) (Gilleland et al, 1995; ughes et al, 1992; Rawling et al, 1995; von Specht et al, 1995). Rawling and her colleagues also demonstrated using overlapping peptide methodology that among 10

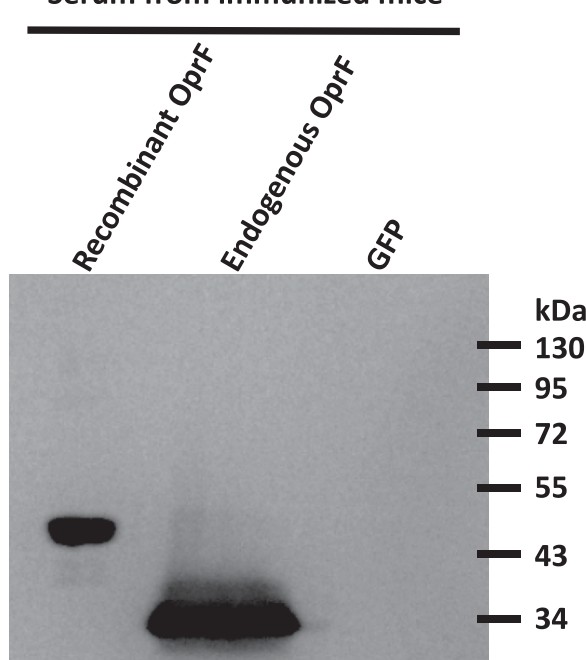

**Serum from immunized mice**

**Figure 10. Western blot illustrating the binding of both endogenous and recombinant OprF by polyclonal antibodies found in sera from immunized mice.**
GFP was used as a negative control. The secondary antibody binding polyclonal antibodies was an HRP-conjugated rabbit anti-mouse IgG and it was detected using chemiluminescence.

surface-reactive OprF-specific antibodies, three of them bound linear epitopes, one belonging to the N-terminal part of OprF (aa 55–62) and two others located in the C-terminal part of the protein (aa 237–244 and 307–314). Whereas the remaining seven recognized conformational epitopes which were assigned to a presumably highly structured and stable two-loop region of 42–90 amino acid long in the middle of the OprF open channel conformer (aa 152–240) (Rawling et al, 1995). The OprF/I fusion protein (containing the C-terminal part of OprF [aa 190–342] fused to the OprI protein [aa 21–83]) of the IC43 vaccine has been produced for 25–30% of the total protein under a soluble form in *E. coli* (Mansouri et al, 1999). This soluble form does not reflect the correct conformation of the OprF full-length protein in *P. aeruginosa* or in the outer membrane vesicles and does not correspond to one of the conformers described by Sugawara and colleagues (Sugawara et al, 2006, 2012). One can suppose that this soluble form may partially reflect the globular conformation of the C-terminal domain found in the periplasm that does not expose native epitopes. However, it is also not clear whether the addition of the OprI amino acids is beneficial for the creation of additional epitopes.

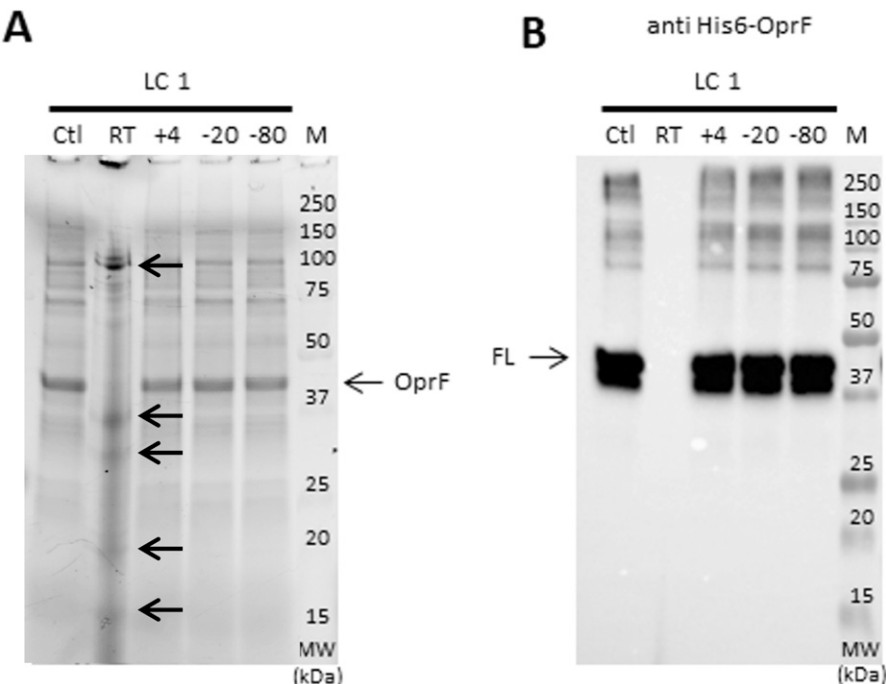

**Figure 11. Analysis of the OprF proteoliposomes at different temperatures. (A, B)** Evaluation of OprF stability in liposomes before (clt, control) and after 14 d storage at RT, 4°C, −20°C, and −80°C using Coomassie gel—SDS–PAGE (A) and anti-HIS Western blot analysis (B). The anti-His antibody was conjugated to HRP and was detected using chemiluminescence.

To the opposite, the second approach used the N-terminal domain of OprF (aa 25–200) instead of its C-terminal part as a potential vaccine candidate. The corresponding Histidine-tagged recombinant N-terminal OprF was expressed in *E. coli* and purified under denaturing conditions. The authors mentioned an additional refolding step that manifestly did not allow the protein to adopt and to stabilize its native eight stranded $\beta$-barrel folding because it did not refer to the use of any detergent nor lipid. Mice were thus at the best immunized against linear epitopes belonging to OprF$_{25-200}$. The authors reported that mice immunization with OprF$_{25-200}$ conferred a maximum protection rate of 50%, 48 h after a challenge with *Acinetobacter baumannii*, a clinical isolate expressing OprF (Bahey-El-Din et al, 2020).

In this study, we reconstituted for the first time the entire OprF membrane antigen in the lipid bilayer of synthetic liposomes by using a cell-free expression system and we tested the vaccine potential of this recombinant proteoliposome exposing all the linear and conformational epitopes of OprF.

OprF cell-free synthesis in the presence of LC1 (cholesterol, DOPC, DOPE, and DMPA; 2-4-2-2) liposomes allowed the obtaining of large amounts of OprF reconstituted into liposomes (protein yield ranging from 0.5 to 1 mg/ml) and the subsequent two steps purification process ended up with highly pure OprF proteoliposomes (≈95% purity). Their analysis by AFM and limited proteolysis demonstrated that most of the OprF proteins were embedded into the liposomal membrane in the correct orientation and in accordance with the two transmembrane topologies described for the closed and open channel conformer of OprF. AFM performed in the presence of Triton X-100 indicated that OprF His-tag was mainly oriented inside the LC1 liposomes and confirmed the existence of two transmembrane topologies resistant to trypsin digestion (Fig 1E). 36% of the protein analyzed were found to span 16 times the

liposomal membrane, which corresponds to the membrane topology of the rare (described previously as <5%) single-domain, open-channel conformer of OprF; whereas 64% of the remaining protein were found to span eight times the liposomal membrane, as described for the membrane topology of the two domains, closed-channel conformer of OprF. By containing a larger proportion of the open-channel conformer of OprF, our proteoliposomes could thus be used as a model to investigate the role of this rare conformation in the different functions allocated to OprF as well as in the immune response mediated by the protein.

In addition, analysis of OprF proteoliposomes by negative staining electron microscopy and by AFM highlighted the formation of large OprF pores in LC1 liposomes. In our conditions, OprF induced the formation of pores with an average diameter of 9.5 ± 4 nm which is larger than the pore diameter (around 0.2 nm) previously allocated to the closed conformation of OprF (Straatsma & Soares, 2009). The formation of such mega pores could be largely the result of OprF oligomerization to a large extent, as suggested by the results of the Western blot experiments demonstrating that OprF forms oligomers which are resistant to chemical and thermal denaturation, as reported by Sugawara et al (2006). OprF dimers, trimers, and tetramers were clearly visible as well as anti-His signals appearing above molecular weight 250 kD, suggesting the existence of OprF higher order assemblies. The fact that pores appear as black holes could also be the consequence of liposomal membrane lysis, induced by an important perforation by OprF. Furthermore, pore distribution at the surface of liposomes was not homogeneous and pores seemed to gather in different areas at the surface of the same liposome. Because such pore formation was not observed with OprF proteoliposomes having a lipid composition close to the one found in Pa membrane (LC2) (data not shown), it remains to determine if this pore formation and in particular, this

large OprF oligomerization, could be promoted by the presence of cholesterol in the liposomal membrane as it is the case for cytolysins (Tweten, 2005). This hypothesis would also explain why pores gathered only within certain areas at the surface of the liposomes, suggesting that these membrane areas were more concentrated in cholesterol. Although *P. aeruginosa* membrane does not contain cholesterol, this is of importance because *P. aeruginosa* cytotoxicity has been related to the presence on host cell membranes of lipids rafts enriched in cholesterol, and OprF was found to be involved in *P. aeruginosa* invasion and evasion events through eukaryotic membranes (Azghani et al, 2002; Grassmé et al, 2003; Fito-Boncompte et al, 2011; Mittal et al, 2016). OprF has notably been found to be implicated in *P. aeruginosa* evasion from macrophages through cell lysis (Garai et al, 2019). One could thus imagine that OprF could use cholesterol of eukaryotic cells to gather and to form mega pores through their membranes, thereby inducing cell disruption.

Regarding the assessment of OprF porin activity, we already reported electrophysiological measurements performed on OprF proteoliposomes fused to a t-BLM with similar conductance (Maccarini et al, 2017). This time we characterized the ion permeability of the OprF porin to potassium (blue symbols) when we added increasing KCl concentrations into the upper part of the t-BLM wells. As shown in Fig S1, we verified that the conductance was proportional to the ln $(K^+)$out (when $(K^+)$in and electric potential are constant) and concluded that the porin was permeable to KCl. In the control well (no protein, green symbols), we noticed a small conductance increase for concentrations of KCl over 1,000 mM, suggesting that only a small increase in permeability of the t-BLM resulted from the very large osmotic difference across the membrane.

Because Wu and colleagues demonstrated that IFN-γ binds OprF and triggers cellular responses enhancing the virulence phenotype of *P. aeruginosa* (Wu et al, 2005), we also investigated the effect of IFN-γ on the ion channel activity of OprF in KCl. We previously described this effect as being transitory and not different to the effect of BSA in NaCl (Maccarini et al, 2017). This time, we were able to observe a permanent inhibitory effect in response to increasing concentrations of IFN-γ. This could be explained by the use of other experimental conditions when performing conductance measurements such as the different lipid compositions of t-BLM as well as by the use of a different model to analyze the conductance measurements. In addition, CD experiments demonstrated that IFN-γ addition to OprF proteoliposomes had an influence on OprF secondary structure and especially, on its predicted β strand percentage going here from 36 to 26%, mainly producing disordered structures. These observations suggest that IFN-γ could destabilize the β barrel structure of the protein, probably through interactions either with the exposed loops of the β barrel domain or with the liposome membrane itself, thus impairing its activity. This last property could be a mechanism allowing the bacteria to escape from the host immune system by blocking its OprF pore.

In summary, we demonstrated that producing OprF full-length protein using a cell-free system in presence of LC1 liposomes allowed fostering the occurrence of its rare open-channel conformer as well as the formation of mega-pores in the liposomal membrane. This open-channel conformer exposed new native and conformational epitopes, which exist neither in the closed channel conformation, nor in the N-terminal or C-terminal part of OprF expressed as a soluble form in previous vaccines. Consequently, OprF proteoliposome may be a better candidate for the building of a vaccine against *P. aeruginosa* and for the development of new treatments against nosocomial and CF infections. To confirm this hypothesis, we performed in vivo vaccination studies in mice using the OprF proteoliposome and analyzed their effect on mice survival. We demonstrated that mice immunization with OprF proteoliposome conferred a 90% protection rate against a lethal dose of the mucoid CF isolate CHA strain. We then analyzed sera from immunized mice and we showed that OprF proteoliposome induced the production of polyclonal antibodies capable of recognizing both the recombinant OprF protein reconstituted into liposomes and the native protein from bacterial lysates, and especially of neutralizing infections in naïve mice because of bacterial challenges after injection of lethal doses of *P. aeruginosa*. The vaccine response obtained with OprF proteoliposome against a mucoid strain of *P. aeruginosa* was thus higher than those reported for the others vaccine strategies based on partial OprF protein, highlighting the advantage of our approach compared with others (Hassan et al, 2018; Knapp et al, 1999; von Specht et al, 1995). Our cell-free membrane antigen expression system in the presence of synthetic liposomes results in a rapid process for obtaining highly effective recombinant vaccines. Our production strategy allows the synthesis of membrane proteins of different complexities and properly folded compared with production of protein fragments or peptides containing linear epitopes. However, our expression system may have some limitations in the production of membrane antigens bearing post translational modifications or of high molecular weights. Finally, we also demonstrated that OprF proteoliposomes can be stored at 4°C, −20°C, and −80°C for at least 2 wk without protein degradation. This is of particular importance because OprF proteoliposomes storage at 4°C would meet the storage conditions necessary for vaccine preservation. In conclusion, we demonstrated that by using a cell-free expression system in the presence of liposomes, we managed to produce, in a one-step-process, a stronger OprF vaccine candidate exposing new epitopes and that does not require to include other immunogens, such as OprI/flagellin B or OprI/PcrV, to be protective (Hassan et al, 2018; Fakoor et al, 2020). As a consequence, our vaccine candidate based on OprF proteoliposomes shows great promise in term of protection responses for intensive care unit patients and brings new hope to have a vaccine against *P. aeruginosa* at our disposal.

# Materials and Methods

### Construction of the *E. coli* recombinant OprF expression vector

The recombinant vector pIVEX2.4-OprF with an N-terminal His-tag was constructed by cloning the PCR-amplified OprF gene from genomic DNA of *P. aeruginosa* (forward primer, 5′-GGAATTCCATATGAAACTGAA-GAACACCTTAG-3′; reverse primer 5′-GTAGAAGCTGAAGCCAAGTAACTCG AGTAACGC-3′) into the expression vector pIVEX2.4d (Roche Diagnostics). The resulting PCR fragment was isolated, purified (QIAquick gel extraction kit, QIAGEN), digested with NdeI and XhoI (Roche Diagnostics)

and ligated (Rapid DNA ligation kit, Roche Diagnostics) into the digested pIVEX2.4d plasmid to obtain the pIVEX2.4-OprF recombinant plasmid, which was checked by sequencing (LGC Genomics).

## Liposomes preparation

Liposomes were prepared by initially drying a lipid mixture in chloroform (Lipid Composition 1 [LC 1]: cholesterol, 1,2-dioleoyl-sn-glycero-3-phosphocholine [DOPC], 1,2-dioleoyl-sn-glycero-3-phosphoethanolamine [DOPE], 1,2-dimyristoyl-sn-glycero-3-phosphate [sodium salt] [DMPA], molar ratio [2-4-2-2]; LC 1': LC 1 + 1 mg/ml monophosphoryl lipid A [MPLA]; LC 2: 1-palmitoyl-2-oleoyl-sn-glycero-3-phosphoethanolamine [POPE], 1-palmitoyl-2-oleoyl-sn-glycero-3-phospho-(1'-rac-glycerol) [sodium salt] [POPG], E. coli cardiolipin [CL], molar ratio [6-2-2]; LC 2': LC 2 + 1 mg/ml MPLA; LC 3: POPE, POPG, E. coli CL, DMPA, molar ratio [6-2-1-1]; LC 3': LC 3 + 1 mg/ml MPLA) (Avanti Polar Lipids) by evaporation under a nitrogen stream. Residual traces of chloroform were removed using a vacuum pump. The lipidic film was hydrated in 500 $\mu$l of Tris buffer (50 mM, pH 7.5) by pipetting and vortexing followed by four freeze/thaw cycles in liquid nitrogen. The lipidic mixture was extruded using an extruder (Avanti Polar Lipids) to produce liposomes with an average diameter of 200 nm. Liposomes were stored at 4°C.

## Cell-free OprF expression in liposomes and purification

The full length OprF from P. aeruginosa was synthetized in presence of liposomes using a CFPS kit (RTS 500 ProteoMaster E. coli HY Kit; Biotechrabbit). The recombinant plasmid pIVEX2.4-OprF and the liposomes of one of the six lipid compositions (LC 1–3') were combined with the cell lysate and the reaction mixture at a final concentration of 15 $\mu$g/ml and 1–4 mg/ml, respectively. Recombinant OprF was produced for 16 h at 25°C and 300 rpm (it's a mixing frequency from the Eppendorf SmartBlock Thermoblock) in a 1.5 ml tube for 50 $\mu$l–1 ml batch. The resulting proteoliposomes were then purified in two steps, a sucrose-gradient centrifugation step and a NaCl washing step. The cell-free reaction mixture was first loaded on top of a 10 ml 0–40% sucrose gradient in Tris buffer (50 mM, pH 7.5) and the tube was centrifuged at 287,660g for 2 h in a TH-641 swinging rotor. 1 ml fractions were recovered from the top to the bottom of the gradient and analyzed by Western blot which was developed with an anti-6-His antibody coupled with HRP (Sigma-Aldrich). Tris buffer (50 mM, pH 7.5) was then added to each fraction containing OprF proteoliposomes and the solution was centrifuged at 30,000g for 30 min at 4°C to pellet proteoliposomes. The pellet was washed twice for 30 min at 4°C with 5 M NaCl and then resuspended in a convenient volume of Tris buffer (50 mM, pH 7.5). The purity and the protein concentration of OprF proteoliposomes were determined on a Coomassie stained SDS–PAGE gel. OprF concentration was obtained by comparison to a range of known BSA concentrations.

## Trypsin digestion

OprF proteoliposomes (LC 1) purified by sucrose gradient centrifugation were proteolyzed using a trypsin:protein mass ratio of 1:10

at RT. Samples were recovered at different times and loaded on a SDS–PAGE gel for a subsequent Western blot analysis.

## CD

CD spectra were recorded with a Jasco J-810 spectropolarimeter at 20°C using a 0.1-cm quartz cell, 10-nm/min scanning speed, and 10-nm bandwidth. OprF concentration in proteoliposomes (LC 1) was 0.8 $\mu$M (0.03 mg/ml) in 10 mM sodium phosphate buffer (pH 7.5). Measurements were performed with OprF proteoliposomes before and after addition of 0.1 mg/ml IFN-γ (Miltenyi Biotec). CD data were blanked against buffer (with or without 0.1 mg/ml IFN-γ) and converted to mean residue ellipticity units. The data were smoothed with a Savitzky–Golay filter using Capito (Wiedemann et al, 2013), a Web server based analysis and plotting tool for CD data and the secondary structure prediction was performed using the Web server BeStSel (Beta Structure Selection) (Micsonai et al, 2018).

## Negative staining electron microscopy

Samples were prepared using the negative stain on grid technique. 10 $\mu$l of OprF proteoliposomes (LC 1, [OprF]: 0.1 mg/ml) or 10 $\mu$l of liposomes (4 mg/ml) in the cell-free reaction mixture without DNA (=negative control) were added to a glow discharge grid coated with a carbon supporting film for 3 min and the grid was stained with 50 $\mu$l of phosphotungstate acid (PTA, at 1% in distilled water) for 2 min. The excess solution was soaked off by a filter paper and the grid was air-dried. The images were taken under low dose conditions (<10 e$^-$/Å$^2$) with defocus values between 1.2 and 2.5 $\mu$m on a Tecnai 12 LaB6 electron microscope at 120 kV accelerating voltage using CCD Camera Gatan Orius 1000. Average pore size was determined using the open source image processing program ImageJ.

## Effect of different storage temperatures on OprF proteoliposomes stability

Samples from the same batch of OprF proteoliposomes (LC 1) were stored at RT, 4°C, –20°C, and –80°C for 2 wk. The integrity of OprF protein in liposomes was evaluated before (control) and after 14 d storage under each condition using SDS–PAGE and Western blot analysis.

## AFM tip functionalization

Gold tips (NPG-10; Bruker Nano AXS) were covered with NTA-self-assembled monolayer after overnight incubation in a solution of NTA-self-assembled monolayer (Prochimia) 0.1 mM in ethanol. Then the tips were rinsed extensively with ethanol, dried with nitrogen and incubated for 1 h in 40 mM NiSO$_4$ in PBS solution and stored at 0–5°C.

## FD-based AFM

A resolve AFM (Bruker) was operated in the "PeakForceTapping" mode. Rectangular cantilevers with nominal spring constants of ≈0.06–0.12 Nm$^{-1}$ and a resonance frequency of ≈18 kHz in water were chosen. All AFM experiments were performed in imaging buffer solution at RT (≈24°C). Adhesion maps were obtained by oscillating the

functionalized tip at 0.25 kHz, with an amplitude of 25 nm, and applying an imaging force of 100 pN. Topographs of 128 × 128 or 256 × 256 pixels were performed scanning 0.125 line/s. The retraction speed was 1,500 nm/s and the contact time between tip and sample was 500 ms.

### Data analysis

The force versus curves of each interaction recognition experiment were saved and exported as text files. NanoScope Analysis v1.9 and BiomecaAnalysis were used to translate force versus time curves into FD curves showing specific adhesion events. The obtained force versus distance curves were then analyzed on the basis of the Worm-Like Chain model. This model is the most suitable and frequently used to describe the extension of polypeptides. The extension z of macromolecule is related to the retraction force $F_{adh}$ via Equation (1):

$$F_{adh}(z) = -\frac{K_B T}{l_p}\left(\frac{z}{l_c} + 4\left(1 - \frac{z}{l_c}\right)^{-2} - \frac{1}{4}\right). \tag{1}$$

Here, the persistence length $l_p$ is a direct measure of the chain stiffness, $l_c$ is the total contour length of the biomacromolecule, and Kb is the Boltzmann constant. The number of monomers in the polypeptide chains was then derived from following Equation (2):

$$N = \frac{l_c}{l_p}. \tag{2}$$

### Incorporation of recombinant OprF into tethered lipid bilayer systems

To evaluate the OprF porin activity, we used a commercially available device, that is, a TethaPod (SDX Tethered Membranes) for the production of t-BLMs. The device comprised six wells that contained a preformed tethering layer made of benzyl disulfide undecaethylene glycol phytanol and benzyl disulfide tetra ethylene glycol polar spacer molecules in a 10:90 ratio. A t-BLM was formed in each of the six wells by addition of a 3 mM ethanolic solution of glycerodiphytanylether:diphytanyletherphosphatidylcholine (30:70 M ratio) (AM199; SDX Tethered Membranes), which self-assembled into a lipid bilayer membrane after being rinsed with the buffer (50 mM Tris pH 7.5, 150 mM KCl). The porin was incorporated into the t-BLMs after addition into the well of 15 $\mu$l of OprF proteoliposome, and 15 $\mu$l of empty liposome was used for the control well. The fusion of OprF proteoliposomes or empty liposomes to the t-BLM was assisted by Dodecyl-$\beta$-Dmaltoside (DDM) used at a concentration below one-fifth of its critical micellar concentration (CMC, 0.01%) to allow the fusion without membrane perturbation. To measure the conductance of pore-forming proteins, the Tethapod system used an alternating-current impedance spectroscopy technique that was adapted to t-BLMs (Cranfield et al, 2014; Siebert et al, 2020). To evaluate K+ transport through the porin, increasing concentrations of KCl (0.15–1 M) were superfused into each well. The TethaPod applied a sequential 20 mV excitation over the 1 kHz–0.1 Hz frequency range. The output measured from the electrical impedance spectroscopy was modelled using the Tethaquick software (v2.0.49; SDX Tethered Membranes) to

determine the capacitance (Cm) of the t-BLM, its resistance (Rm), and a constant phase element accounting for the impedance of the tethered region below the t-BLM (CPEteth). During the experiments, each t-BLM remained stable, as shown by the low variability of the t-BLM capacitance before porin incorporation (10.9 ± 0.4 nF, mean ± SD, n = 25). The conductances are the mean of three conductances measured during 10 min for each KCl concentration (Fig S1A) and for each IFN-γ concentration (diluted in 150 mM KCl) (Fig S1B).

### Animal vaccination protocols and ethics statement

Mice were purchased from Janvier SA and were kept under specific pathogen-free conditions. Animal experiments were performed in accordance with the institutional and national guidelines approved by Animal Experiment Committee of the Region and Use Committee of Grenoble Alpes University (UGA). 6 wk old female C57Bl/6J were anesthetized (isoflurane) and before immunization, a retro-orbital blood sample was collected on each mouse and sera conserved at −80°C. Mice were immunized subcutaneously four times at 2-wk intervals by administering 25, 50, or 75 $\mu$g of recombinant OprF proteoliposome. A control group received liposome from the same composition as for OprF and which was previously incubated with cell-free lysate. A blood sample from retro-orbital was obtained from all mice 1 wk after the second and the third immunization. A final blood sample was obtained 5 d after the bacterial challenge. 1 wk after the fourth immunization, mice were subcutaneously infected with a lethal dose of CHA P. aeruginosa ($2 × 10^7$ CFU/mouse). After infection, all mice were daily monitored for clinical signs including weight loss and behavior. Mice survival was checked every 4 h for up to 80 h. Animal health was recorded and mice were euthanized if necessary. To ensure the reproducibility of the vaccine scheme, two sets of vaccination experiments were performed using different OprF proteoliposome preparations and used at 25, 50, or 75 $\mu$g for the immunization protocols.

Cross protection experiments were performed by challenging mice with a lethal dose of CHA P. aeruginosa injected subcutaneously ($2 × 10^7$ CFU/mouse) and 1 h after infection, mice were injected intraperitoneally either with 100 ml of sera from negative control or 100 ml of pooled sera from vaccinated mice. After infection, all mice were daily monitored for clinical signs including weight loss and behavior. Mice survival was checked every 4 h for up to 80 h. Animal health was recorded and mice were euthanized if necessary.

### Bacterial strains and culture methods

P. aeruginosa strain used was CHA isolated from a patient at the Grenoble hospital (Bezuidt et al, 2013). For mice infection, P. aeruginosa CHA was grown in Luria Broth (Sigma-Aldrich) overnight at 37°C, 150 rpm (it's the rotation speed of the incubation shaker). For mice injections, bacteria were washed twice with PBS and resuspended in PBS at $2 × 10^8$ CFU/ml. Mice were injected subcutaneously with 100 $\mu$l of CHA at ~$2 × 10^7$ CFU/mouse.

### ELISA

Serum samples from mice immunized with 25, 50, or 75 $\mu$g of recombinant OprF proteoliposome or with liposome (negative control)

were obtained from retro-orbital samples 1 wk after the fourth immunization (day 48) for the ELISA. Wells of microtiter plates (Sigma-Aldrich) were coated with OprF proteoliposomes or liposomes (1,000 ng per well) in 0.1 M carbonate buffer (pH 9.5) overnight at 4°C. Diluted serum samples were used as the primary antibodies. The secondary antibody was HRP-conjugated rabbit anti-mouse IgG. The optical density at 450 nm was measured, and the titers were defined as the highest dilution that yielded an absorbance value of more than twice the value of the control serum.

### Preparation of endogenous OprF

2 ml of an overnight culture of *P. aeruginosa* CHA strain were centrifuged to pellet bacteria. This cell pellet was then resuspended in blue loading buffer and boiled for 10 min before loading on a SDS–PAGE gel for subsequent Western blotting.

## Supplementary Information

## Acknowledgements

The authors thank Hélène Coradin for her technical assistance, Dalil Hannani, Bertrand Toussaint, Thomas Soranzo, and Audrey Le Gouellec for their helpful advices. The EM work was done using the platforms of the Grenoble Instruct-ERIC Center (ISBG : UMS 3518 CNRS-CEA-UGA-EMBL) with support from FRISBI (ANR-10 INBS-05-02) and GRAL (ANR-10-LABX-49-01) within the Grenoble Partnership for Structural Biology (PSB). The electron microscope facility is supported by the Rhône-Alpes Region, the Fondation Recherche Medicale, the fonds FEDER, the Centre National de la Recherche Scientifique (CNRS), the CEA, the University of Grenoble, European Molecular Biology Laboratory (EMBL), and the GIS-Infrastrutures en Biologie Sante et Agronomie (IBISA). The Plateforme Chimie Nanobio de l'Institut de Chimie Moléculaire (PCN-ICMG) (FR 2607) is acknowledged for providing facilities for circular dichroism experiments (M Jourdan). The authors thank for funding the research presented in this manuscript: the European Commission for a grant from a Marie Curie Excellence Grant (MEXT-014320, FP6, European Commission), and the Sociétés d'Accélération du Transfert de Technologies (SATT) Linksium for a grant (Grenoble, France). We would like to also thank Charlotte Lombardi for her helpful advices.

### Author Contributions

G Mayeux: conceptualization, validation, investigation, visualization, methodology, and writing—original draft, review, and editing.
L Gayet: investigation.
L Liguori: investigation.
M Odier: investigation.
DK Martin: resources.
S Cortès: investigation.
B Schaack: validation, investigation, visualization, methodology, and writing—original draft.
J-L Lenormand: conceptualization, resources, data curation, funding acquisition, validation, visualization, methodology, project administration, and writing—original draft, review, and editing.

### Conflict of Interest Statement

The authors declare that the research was conducted in the absence of any commercial or financial relationships that could be construed as a potential conflict of interest.

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
