## [Reviewer comments · Life Science Alliance]

Life Science Alliance

Cell-free expression of the outer membrane protein OprF of *P. aeruginosa* for vaccine purposes

Géraldine Mayeux, Landry Gayet, Lavinia Liguori, Marine Odier, Donald Martin, Sandra Cortés, Béatrice Schaack, and Jean-Luc Lenormand

DOI: <https://doi.org/10.26508/lsa.202000958>

Corresponding author(s): Jean-Luc Lenormand, University Grenoble Alpes

Review Timeline:

Submission Date:	2020-11-14
Editorial Decision:	2021-03-25
Revision Received:	2021-04-22
Accepted:	2021-04-26

Scientific Editor: Shachi Bhatt

Transaction Report:

March 25, 2021

RE: Life Science Alliance Manuscript #LSA-2020-00958-T

Prof. Jean-Luc Lenormand
University Grenoble Alpes
School of Pharmacy
Building Jean Roget
Domaine de la Merci
La Tronche 38700
France

Dear Dr. Lenormand,

Thank you for submitting your revised manuscript entitled "Cell-free expression of the outer membrane protein OprF of *P. aeruginosa* for vaccine purposes".

We apologize for this extended and unusual delay in getting back to you. As you will note from the reviewers' comments below, the reviewers are quite enthusiastic about these findings and have asked for only minor text revisions to be made. We would thus accept this manuscript with minor revisions and encourage you to submit a revised version back to us that addresses these reviewers' points and that is revised in accordance to LSA's formatting guidelines.

Along with the points listed below, please also attend to the following:

- please upload your main and supplementary figures as single files
- please add your main and supplementary figure legends to the main manuscript text after the references section
- please use the [10 author names, et al.] format in your references (i.e. limit the author names to the first 10)
- please rename Figure S3 to Figure S2, both in its label and in callouts in the manuscript text, otherwise, Figure S2 is missing
- please consult our manuscript preparation guidelines <https://www.life-science-alliance.org/manuscript-prep> and make sure your manuscript sections are in the correct order;
- please add Keywords, a Category, and a Summary Blurb/Alternate Abstract for your manuscript in our system
- please add an Author Contributions section to your main manuscript text and in the system
- please add callouts for Figures 6E; 7A-C; 11A-B to your main manuscript text
- Figures containing gel images have resolution problems that need to be fixed in the revised version.

A. FINAL FILES:

B. MANUSCRIPT ORGANIZATION AND FORMATTING:

Sincerely,

Shachi Bhatt, Ph.D.

Executive Editor

Life Science Alliance

<https://www.lsjournal.org/>

Interested in an editorial career? EMBO Solutions is hiring a Scientific Editor to join the international Life Science Alliance team. Find out more here -

https://www.embo.org/documents/jobs/Vacancy_Notice_Scientific_editor_LSA.pdf

Reviewer #1 (Comments to the Authors (Required)):

General comments for authors:

"Pseudomonas aeruginosa is the second-leading cause of nosocomial infections and pneumonia in hospitals. Due to its extraordinary capacity for developing resistance to antibiotics, treating infections by Pseudomonas is becoming a challenge, lengthening hospital stays, and increasing medical costs and mortality." To address the need for a vaccine Dr. Mayeux and co-authors developed a cell-free approach to produce the outer membrane protein OprF from P. aeruginosa using proteoliposomes that appear to mimic the natural lipid environment for the membrane protein. Extensive biophysical measurements such as AFM and circular dichroism to assess the protein structure and pore forming capabilities. This approach has been shown by others to be feasible for vaccine development and the authors do a good job of extending this idea to the OprF from P. aeruginosa. The overall data appears to be compelling. Statistical analysis and in-depth measurement of the IgG and IgA responses would have increased the scientific impact of the study. Overall, the manuscript is well written and an excellent example of qualifying the parameters membrane-bound proteins for use in immunological studies. I support the publication of the manuscript with the consideration of addressing Specific Comments below.

Minor specific comments for authors:

1. For the protein production and functional as well as structural characterization the data is strongly supportive.
2. The manuscript lacked an objective view regarding some of the limits of the study. The immunological studies were very compelling, but how does the proteoliposomes containing recombinant OprF compare to wild-type was not directly addressed within the discussion of the manuscript. The pros and cons of the study would be helpful to the readers.
3. Authors need to carefully double check and ensure random abbreviations are defined.
4. The authors refer to the lipid mixes as providing a native environment, when the term near-native is more accurate.
5. A table of the lipid mixtures and net protein yields would be helpful and could be combined with

Figure 1.

6. Figure 1., could be shortened by focusing on panels C and D. The remaining images could go to the supplemental section.

7. Figure 6. Can be limited to the first 2 panels. Contrast in Panels B-E appear to be image enhanced compared to the control?

8. For the immunological studies, the data supports the authors results, but additional information will be needed regarding number of mice, replicate experimental numbers with prime boost scheduling.

9. An overall description of the statistical methods was lacking from the Materials and Methods section.

Reviewer #2 (Comments to the Authors (Required)):

This study demonstrates the possibility to produce in vitro massive amounts of the His-tagged OprF major outer membrane protein of *Pseudomonas aeruginosa* and insert it in proteoliposomes. The resulting proteoliposomes were further separated from other fractions using sucrose gradient centrifugation. Using AFM and trypsin digestion the authors demonstrate that OprF is either fully embedded and protected or 1/2 embedded and trypsin susceptible, corresponding to close (8 TM) and known rare open channel with 16 TM (normally <5% but here, interestingly up to 36%). An other very interesting observation is the effect of IFN-gamma addition causing a pore reduction as well as reduced conductance. Using EM and AFM, pores of 9.5 nm could be confirmed, another very striking result. Finally, the OprF proteoliposomes were used to vaccinate mice which were further challenged with a CF mucoid isolate of *P. aeruginosa*. Full protection was observed and injection of polyclonal anti-OprF antibodies also conferred protection. The humoral response was also found by ELISA to be very strong. Last and not least, the OprF proteoliposomes were found to be stable at 4°C for 14 days.

Remarks and comments:

- Please insert line numbers, especially after revision.
- It is a pity that one has to wait to read the methods at the end of the manuscript to get some details (such as abbreviations, which His-tag, etc.). This should also be mentioned in the results section or the methods section should be moved at the beginning of the manuscript.
- Some figures of gels are of poor quality, please increase the definition (ex: Figure 1).
- Please define "abortive synthesis", premature termination of translation?
- Why the choice of a mucoid CF isolate inoculated sub-cutaneously? is it because the alginate layer makes the OprF exposure less likely and the vaccination more challenging? Did you try with another non-mucoid isolate?
- In the Discussion the part concerning the effect of IFN-gamma shows a contradiction since you mention that IFN-gamma increases the virulence and later on that it probably causes a decrease in virulence.
- Also, please elaborate on the possible vaccination strategies using your proteoliposomes, especially in the case of chronic cF infections.

Reviewer #3 (Comments to the Authors (Required)):

In this work, the authors seek to study the vaccine properties of a proteoliposome containing the major outer membrane protein OprF of *Pseudomonas aeruginosa*. This new strategy constitutes a relevant lead in the field of anti-pseudomonas control.

To do this, the authors show that the proteoliposomes can be produced by an *E. coli* cell-free expression system and purified on a sucrose gradient (Part I). In the second part of the article, devoted to the biochemical characterization of proteoliposomes, the authors sought information on the orientation and topology of the protein in the membrane. Interestingly, the authors observe Fig3, the appearance of a band corresponding to approximately half of the recombinant protein and attribute to the existence of 2 potential conformations incorporated into the proteoliposomes. It might be of interest to characterize peptide mass spectrometry to identify the protein fragment that remains protected in membranes from tryptic digestion.

It has been shown previously that IFN- γ binds to OprF and induces a signaling cascade (Wu et al., 2005). The authors show in this article that the addition of IFN- γ influences the topology of the protein incorporated into the proteoliposomes as well as the conductance of the channel. It would also be interesting to know if the presence of the pores observed in ME is also influenced by the presence of IFN- γ . On the other hand, how can the authors say that the observed pores are surrounded by OprF proteins? Staining with antibodies should allow them to confirm this. The article is very well structured, well written and very pleasant to read. The quality experimental results are well described and the necessary context is given to the reader by citing the existing literature.

However, it would be desirable to review the typography of the reference list.

RESPONSES TO REVIEWERS' COMMENTS**Life Science Alliance #LSA-2020-00958-T**

Title: Cell-free expression of the outer membrane protein OprF of *Pseudomonas aeruginosa* for vaccine purposes.

Authors: Géraldine Mayeux, Landry Gayet, Lavinia Liguori, Marine Odier, Donald K. Martin, Sandra Cortès, Béatrice Schaack and Jean-Luc Lenormand.

REVIEWER #1 COMMENT #1

General comments for authors:

"*Pseudomonas aeruginosa* is the second-leading cause of nosocomial infections and pneumonia in hospitals. Due to its extraordinary capacity for developing resistance to antibiotics, treating infections by *Pseudomonas* is becoming a challenge, lengthening hospital stays, and increasing medical costs and mortality." To address the need for a vaccine Dr. Mayeux and co-authors developed a cell-free approach to produce the outer membrane protein OprF from *P. aeruginosa* using proteoliposomes that appear to mimic the natural lipid environment for the membrane protein. Extensive biophysical measurements such as AFM and circular dichroism to assess the protein structure and pore forming capabilities. This approach has been shown by others to be feasible for vaccine development and the authors do a good job of extending this idea to the OprF from *P. aeruginosa*. The overall data appears to be compelling. Statistical analysis and in-depth measurement of the IgG and IgA responses would have increased the scientific impact of the study. Overall, the manuscript is well written and an excellent example of qualifying the parameters membrane-bound proteins for use in immunological studies. I support the publication of the manuscript with the consideration of addressing Specific Comments below.

RESPONSE

We thank the reviewer for his/her general comment and for an accurate assessment of the key findings in our manuscript. We also thank the reviewer for supporting the publication of our manuscript. We hope that the reviewer will favorably consider the modifications in the revised version.

REVIEWER #1 COMMENT #2

Minor specific comments for authors:

1. For the protein production and functional as well as structural characterization the data is strongly supportive.
2. The manuscript lacked an objective view regarding some of the limits of the study. The immunological studies were very compelling, but how does the proteoliposomes containing recombinant OprF compare to wild-type was not directly addressed within the discussion of the manuscript. The pros and cons of the study would be helpful to the readers.

RESPONSE

The reviewer raised a valid point concerning the immunological studies using recombinant proteoliposomes as delivery systems for OprF. As suggested by the reviewer, we added some pros and cons of our technology for obtaining recombinant vaccines in the discussion from line 659 starting at "Our cell-free membrane antigen...." To line 665 "...of high molecular weights."

REVIEWER #1 COMMENT #3

3. Authors need to carefully double check and ensure random abbreviations are defined.

RESPONSE

As suggested by the reviewer, we defined every single abbreviation in the text.

REVIEWER #1 COMMENT #4

4. The authors refer to the lipid mixes as providing a native environment, when the term near-native is more accurate.

RESPONSE

We agree with the reviewer and we modified in the text the term “native environment” by “near-native”.

REVIEWER #1 COMMENT #5

5. A table of the lipid mixtures and net protein yields would be helpful and could be combined with Figure 1.

RESPONSE

A table has been added at lane 1239.

REVIEWER #1 COMMENT #6

6. Figure 1., could be shortened by focusing on panels C and D. The remaining images could go to the supplemental section.

RESPONSE

We thank the reviewer for this suggestion but we think that for a better understanding for the reader of the problematic and to show a complete and full story of how we performed the selection of the lipid composition for further experiments, we prefer to show the whole panels as previously.

REVIEWER #1 COMMENT #7

7. Figure 6. Can be limited to the first 2 panels. Contrast in Panels B-E appear to be image enhanced compared to the control?

RESPONSE

We chose to show several EM images of proteoliposomes with pores on their surface first, in order to report that they were observed on a large number of images of the sample and then, in order to be able to appreciate the size as well as the distribution of these pores on the surface of a small number of proteoliposomes.

The difference in contrast between the image of the control liposomes (panel A) and those of the proteoliposomes (panels B-E) is explained by the fact that, unlike the OprF proteoliposomes which

have been purified, the control liposomes have been voluntarily placed in the cell-free reaction mixture in order to demonstrate that the formation of pores through the proteoliposomes is not due to the presence of the other constituents of the cell-free reaction mixture used to express OprF.

REVIEWER #1 COMMENT #8

8. For the immunological studies, the data supports the authors results, but additional information will be needed regarding number of mice, replicate experimental numbers with prime boost scheduling.

RESPONSE

The number of mice used in the different experiments is indicated into the legend of the figure 6. A description of the protocol is also indicated into the legend of the figure: "C57Bl/6 mice were immunized subcutaneously with 25 µg, 50 µg or 75 µg of OprF in proteoliposome. One week after the fourth immunization, OprF-immunized groups (n=10 or 11) or control group (n=11) were subcutaneously challenged with a lethal dosage of *P. aeruginosa* CHA strain ($2 \cdot 10^7$ CFU/mouse) and monitored for survival and disease symptoms for up to 80 hours". These experiments have been performed three times with identical results. The graph is a representation of one of these experiments.

REVIEWER #2 COMMENT #1

This study demonstrates the possibility to produce in vitro massive amounts of the His-tagged OprF major outer membrane protein of *Pseudomonas aeruginosa* and insert it in proteoliposomes. The resulting proteoliposomes were further separated from other fractions using sucrose gradient centrifugation. Using AFM and trypsin digestion the authors demonstrate that OprF is either fully embedded and protected or 1/2 embedded and trypsin susceptible, corresponding to close (8 TM) and known rare open channel with 16 TM (normally <5% but here, interestingly up to 36%). Another very interesting observation is the effect of IFN-gamma addition causing a pore reduction as well as reduced conductance. Using EM and AFM, pores of 9.5 nm could be confirmed, another very striking result. Finally, the OprF proteoliposomes were used to vaccinate mice which were further challenged with a CF mucoid isolate of *P. aeruginosa*. Full protection was observed and injection of polyclonal anti-OprF antibodies also conferred protection. The humoral response was also found by ELISA to be very strong. Last and not least, the OprF proteoliposomes were found to be stable at 4°C for 14 days.

RESPONSE

We thank the reviewer for providing a thoughtful review and for his/her comments on our manuscript.

REVIEWER #2 COMMENT #2

Remarks and comments:

- Please insert line numbers, especially after revision.

- It is a pity that one has to wait to read the methods at the end of the manuscript to get some details (such as abbreviations, which His-tag, etc..). This should also be mentioned in the results section or the methods section should be moved at the beginning of the manuscript.

RESPONSE

As suggested by the reviewer, we introduced the line numbers in the manuscript. Regarding the abbreviations and others details as mentioned by the reviewers, we modified the text in order to clarify some points.

REVIEWER #2 COMMENT #3

- Some figures of gels are of poor quality, please increase the definition (ex: Figure 1).

RESPONSE

We modified the figures to increase the quality of each figure as recommended by the reviewer.

REVIEWER #2 COMMENT #4

- Please define "abortive synthesis", premature termination of translation?

RESPONSE

As suggested by the reviewer, we changed the term "abortive synthesis" by "premature termination of translation" (line 221).

REVIEWER #2 COMMENT #5

- Why the choice of a mucoid CF isolate inoculated sub-cutaneously? is it because the alginate layer makes the OprF exposure less likely and the vaccination more challenging? Did you try with another non-mucoid isolate?

RESPONSE

We thank the Reviewer for raising a valid point concerning the use of a mucoid *Pseudomonas aeruginosa* strain for testing our vaccination protocol. We choose this mucoid PA strain because our first vaccination goal was to develop a recombinant vaccine for patients early diagnosed with cystic fibrosis. We know that OprF is one of the main membrane antigens responsible for the virulence process in these patients. We didn't try non-mucoid PA strains for the moment for testing our vaccination strategy but we will plan to perform these in further experiments.

REVIEWER #2 COMMENT #6

- In the Discussion the part concerning the effect of IFN-gamma shows a contradiction since you mention that IFN-gamma increases the virulence and later on that it probably causes a decrease in virulence.

RESPONSE

We thank the reviewer for his/her comment on the discrepancy in the discussion on the binding of INF- γ to OprF. As described in several studies (e.g. Wu L. et al., 2005; Wagner VE et al., 2006) on the roles of the binding of OprF to interferon-gamma, especially on the upregulation of the production of virulence factors such as LecA lectin through the quorum sensing (QS) system, we decided to remove in the discussion, the sentence "It could also explain the decrease in *P. aeruginosa* virulence observed in chronic infection when IFN- γ concentration peaks." to avoid any confusing purpose.

REVIEWER #2 COMMENT #7

- Also, please elaborate on the possible vaccination strategies using your proteoliposomes, especially in the case of chronic cF infections.

RESPONSE

Although the aim of this study was not to design a vaccination strategy for chronic cystic fibrosis infections, it is conceivable to vaccinate children diagnosed early and before they develop recurrent respiratory infections. This vaccination strategy could be recommended or suggested to parents with CFTR mutations who have newborns.

REVIEWER #3 COMMENT #1

In this work, the authors seek to study the vaccine properties of a proteoliposome containing the major outer membrane protein OprF of *Pseudomonas aeruginosa*. This new strategy constitutes a relevant lead in the field of anti-pseudomonas control.

To do this, the authors show that the proteoliposomes can be produced by an *E.coli* cell-free expression system and purified on a sucrose gradient (Part I). In the second part of the article, devoted to the biochemical characterization of proteoliposomes, the authors sought information on the orientation and topology of the protein in the membrane. Interestingly, the authors observe Fig3, the appearance of a band corresponding to approximately half of the recombinant protein and attribute to the existence of 2 potential conformations incorporated into the proteoliposomes. It might be of interest to characterize peptide mass spectrometry to identify the protein fragment that remains protected in membranes from tryptic digestion.

RESPONSE

We thank the reviewer for providing a thoughtful review and for his/her comments on our manuscript and on our experiments by stating that "This new strategy constitutes a relevant lead in the field of anti-pseudomonas control."

REVIEWER #3 COMMENT #2

It has been shown previously that IFN-g binds to OprF and induces a signaling cascade (Wu et al., 2005). The authors show in this article that the addition of IFN-g influences the topology of the protein incorporated into the proteoliposomes as well as the conductance of the channel. It would also be interesting to know if the presence of the pores observed in ME is also influenced by the presence of IFN-g. On the other hand, how can the authors say that the observed pores are surrounded by OprF proteins? Staining with antibodies should allow them to confirm this.

RESPONSE

It would indeed be very interesting to investigate by electron microscopy the effect of gamma interferon on the presence of pores within proteoliposomes. This is an experiment that we plan to implement very soon.

The presence of OprF around the pores was demonstrated by AFM. Figure 7B shows the presence of specific adhesion forces between the tris-NTA modified tip and OprF His-tag around the pores highlighted by circles in white dotted lines.

Regarding the labeling of OprF with an antibody, one possibility would have been to use an anti-His antibody coupled to gold beads. However, since the protein's His-tag is predominantly oriented

towards the interior of the proteoliposome, it would not have been accessible for binding by the antibody. We could have used a commercial anti-OprF antibody but there is no assurance that this commercial antibody can bind to the open form of the protein responsible for pore formation nor that it can be clearly visualized on the proteoliposome surface by electron microscopy due to its small size.

REVIEWER #3 COMMENT #3

The article is very well structured, well written and very pleasant to read. The quality experimental results are well described and the necessary context is given to the reader by citing the existing literature.

However, it would be desirable to review the typography of the reference list.

RESPONSE

We thank the reviewer for his/her general comment on our manuscript and for allowing to publish it like this. We hope that the reviewer will favorably consider the modifications in the revised version.

April 26, 2021

RE: Life Science Alliance Manuscript #LSA-2020-00958-TR

Prof. Jean-Luc Lenormand
University Grenoble Alpes
School of Pharmacy
Building Jean Roget
Domaine de la Merci
La Tronche 38700
France

Dear Dr. Lenormand,

Thank you for submitting your Research Article entitled "Cell-free expression of the outer membrane protein OprF of *P. aeruginosa* for vaccine purposes". It is a pleasure to let you know that your manuscript is now accepted for publication in Life Science Alliance. Congratulations on this interesting work.

*****IMPORTANT:** If you will be unreachable at any time, please provide us with the email address of an alternate author. Failure to respond to routine queries may lead to unavoidable delays in publication.*******

DISTRIBUTION OF MATERIALS:

Again, congratulations on a very nice paper. I hope you found the review process to be constructive and are pleased with how the manuscript was handled editorially. We look forward to future exciting

submissions from your lab.

Sincerely,

Shachi Bhatt, Ph.D.

Executive Editor

Life Science Alliance

<http://www.lsjournal.org>
